# TopoAlign: A Framework for Aligning Code to Math via Topological Decomposition

## Abstract

Large Language Models (LLMs) excel at both informal and formal (e.g. Lean 4) mathematical reasoning but still struggle with *autoformalisation*, the task of transforming informal into formal mathematical statements. Autoformalisation helps pair the informal reasoning of LLMs with formal proof assistants which enable machine-verifiable generation and mitigate hallucinations. Yet, the performance of current Math LLMs is constrained by the scarcity of large-scale corpora, particularly those containing pairs of informal and formal statements. Although current models are trained to generate code from natural language instructions, structural and syntactic differences between these and formal mathematics limit effective transfer learning. We propose *TopoAlign*, a framework that unlocks widely available code repositories as training resources for Math LLMs. TopoAlign decomposes code into docstrings, main functions, and dependency functions, and reassembles these components into analogues that structurally mirror formal statements. This produces structurally aligned code data that can be used for training Math LLMs without requiring additional human annotation. We train two state-of-the-art models, DeepSeek-Math and Herald, and evaluate them on the MiniF2F, Putnam, and ProofNet benchmarks. TopoAlign provides substantial gains for DeepSeek-Math, improving performance by 17.77% on BEq@10 and 68.82% on typecheck@10. Despite introducing no new mathematical knowledge, our framework achieves gains of 0.12% and 1.09% for Herald on BEq@10 and typecheck@10, respectively, demonstrating that training on aligned code data is beneficial even for specialized models.

## 1 Introduction

Neuro-symbolic approaches that pair Large Language Models (LLMs) with proof assistants, such as Isabelle (Nipkow et al., 2002b) or Lean 4 (Moura & Ullrich, 2021), enable advanced mathematical reasoning by enforcing rule-based logical consistency (Welleck & Saha, 2023). These assistants operate on Formal Languages (FL), such as Lean 4, which provide rigorous, machine-verifiable frameworks. However, proficiency in these formal languages requires specialized expertise, meaning most mathematical problems are initially expressed in Natural Language (NL). While NL is ideal for human communication, its inherent flexibility and contextual dependence make it challenging to translate into a formal system. Bridging this gap requires *autoformalisation*, the process of faithfully translating informal NL math problems into FL. This step is essential for interacting with automated verifiers for tasks such as proof generation (Wu et al., 2022a; Ahn et al., 2024).

Despite recent advances, LLMs still struggle with autoformalisation, in part due to the lack of large-scale, high-quality, parallel datasets that pair NL problem descriptions with corresponding formal statements or proofs (Wu et al., 2022a). Synthetic datasets such as Herald statements (Gao et al., 2025) address the lack of training corpora, but their scale and diversity remain limited—especially compared to domains like code generation, where vast corpora are readily available. As a result, current models often either fail outright or require thousands of attempts and auxiliary retrieval systems to produce accurate formalisations of even simple mathematical problems (Li et al., 2024).

We address this bottleneck by extending the training resources available for Math LLMs to include widely available code repositories. Recent work demonstrates that models can learn the structure

---

†Equal contribution.

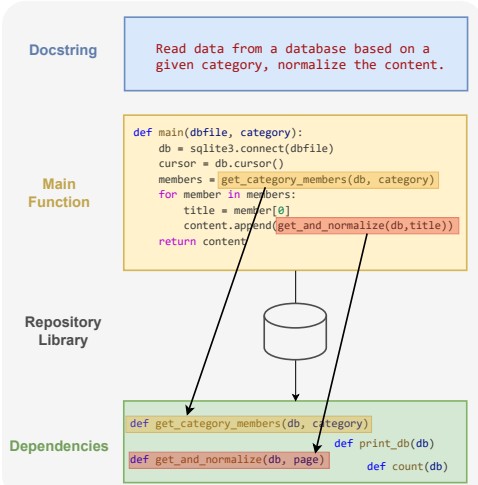 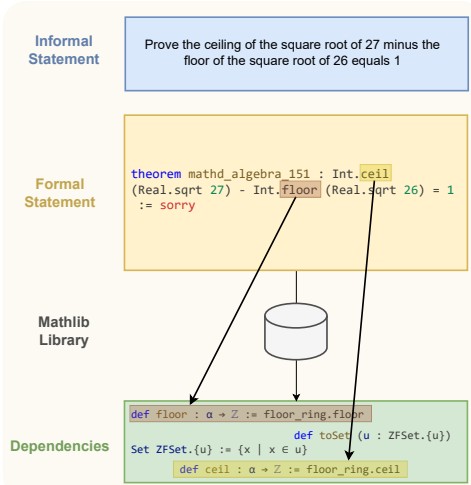

Figure 1: Structural similarity between code (left) and formal statements in Lean 4 (right). Code samples extracted from GitHub repositories are decomposed into: the docstring, which maps to informal statements in mathematical problems, the main function, which corresponds to the formal statements, and the dependency functions, which correspond to supporting lemmata and theorems, included in external libraries (e.g. Mathlib for Lean 4).

of a task from syntactically aligned data, even if the data is semantically unrelated to the final task (Gandhi et al., 2024). This suggests that vast programming code corpora could be leveraged to teach the compositional patterns of formal mathematics, provided the structure is correctly aligned. To achieve this, we propose *TopoAlign*, a framework that structurally aligns programming code with formal mathematics. TopoAlign decomposes code into docstrings, main functions, and dependency functions, and reassembles these components into sequences that mirror the structure of Lean 4 formal statements, see Figure 1. This alignment teaches the model the compositional structure of formal mathematics and enables transfer of problem-solving capabilities learned from code without introducing new mathematical knowledge. Applying TopoAlign, we construct a combined corpus of aligned code and formal math data. On top of this corpus, we introduce *code autoformalisation* (CAF), a task that emulates autoformalisation using the aligned code data. Specifically, we align code docstrings, dependency functions and main function bodies with informal descriptions, supporting lemmata, and formal statements in Lean code. Unlike regular code generation, where the challenge consists of solving the problem statement, our setting provides a synthetic docstring that already includes the solution intent, making the task closer to translating an informal mathematical description into a formal statement.

We train DEEPSEEK-MATH (Shao et al., 2024) and HERALD (Gao et al., 2025) with TopoAlign and the CAF objective, and evaluate on the MiniF2F, Putnam, and ProofNet benchmarks. The method yields consistent gains, achieving relative BEq improvements of 36.7% for DEEPSEEK-MATH and 6.2% for HERALD.

**Contributions:. 1)** We introduce TopoAlign, a novel method addressing the shortage of training corpora for Math LLMs by structurally aligning code data with formal mathematical languages. **2)** We propose "code autoformalisation" (CAF), a training task that leverages the structurally aligned code dataset to emulate autoformalisation, thereby reducing the dependence on annotated pairs of informal and formal mathematical statements. **3)** We release a large-scale semi-synthetic pre-training dataset of 300 million tokens, consisting of high-quality, structurally aligned code designed for autoformalisation tasks. **4)** Through detailed ablation studies, we demonstrate that a balanced ratio of our aligned code data and formal mathematical statements yields optimal autoformalisation performance.

## 2 RELATED WORK

Autoformalisation refers to the translation of informal mathematical problems in NL to FL statements. This requires extensive mathematical knowledge and comprehensive understanding of of the problem statements. Autoformalisation is a foundational component for integrating LLMs in neuro-symbolic approaches for tasks like theorem proving (Wu et al., 2022a). This forms a positive feedback loop, as improvements in theorem proving have also been found to enhance autoformalisation (Tarrach et al., 2024). Therefore, advancing autoformalisation is essential for neuro-symbolic approaches and mathematical reasoning.

Previous methods for autoformalisation draw inspiration from machine translation literature (Wang et al., 2018; Dwivedi et al., 2022), i.e. Szegedy (2020) propose encoding NL and FL in a shared latent space and selecting translation candidates based on embedding similarity. Some approaches focus on rule-based methods, such as GFLean, which uses the Grammatical Framework for parsing and linearization (Pathak, 2024). However, these methods struggle to adapt to diverse inputs, as their rules require frequent updates. In contrast, LLMs provide more flexibility and consequently show strong autoformalisation performance (Jiang et al., 2022; Jiang, 2024; Wu et al., 2022b).

Despite their success in narrow domains (Soroco et al., 2025; Zhu et al., 2024), these methods face a common challenge: the scarcity of parallel NL-FL math datasets. Various approaches are aimed to extend the training datasets: ATLAS (Liu et al., 2025b) proposes using a student-teacher model to generate additional synthetic data, but its effectiveness relies on an excellent teacher model, whereas it uses DeepSeek, which the general-purpose teacher reaches a mathematical knowledge boundary. Additionally, ATLAS generates fully synthetic mathematical problems and formalizations from existing Lean code. In contrast, TopoAlign structurally aligns code repositories with formal mathematics and does not introduce new Lean statements. Herald Statements (Gao et al., 2025) are synthetically generated and, of lower quality compared to human-annotated data as they contain variations of existing data. Jiang et al. (2024) show that multilingual data improves autoformalisation performance. Importantly, Chan et al. (2025) highlight that high-quality data can yield further performance improvements. To address this, we leverage structurally aligned code data for training Math LLMs. This provides a scalable alternative to mathematical statements in FL.

Codex demonstrated the power of pretraining on code data, as it achieves noticeable few-shot performance for autoformalisation tasks (Chen et al., 2021). As such, typically, Math LLMs are initialised from LLMs trained on extensive code data and progressively fine-tuned on mathematical datasets. For example, Llemma (Zhang et al., 2024), Kimina (Wang et al., 2025) and DEEPSEEK-MATH (Shao et al., 2024) are commonly trained on code and fine-tuned on math corpora problem. Li et al. (2024) claim that the autoformalisation capabilities of Math LLMs has not been fully exploited using general-purpose code data during pretraining. To address this, we propose using widely available code repositories as a additional sources for Math LLMs by topologically decomposing and aligning code with formal mathematical statements.

Prior work has explored synthetic data generation methods to address the scarcity of autoformalisation data. Approaches, such as ATLAS, propose a student-teacher framework to create new samples, but their effectiveness is capped by the performance of the initial teacher model (Liu et al., 2025b). Other methods, like the Herald Statements dataset, generate variations of existing statements from libraries like Mathlib, which may limit the novelty of the resulting data (Gao et al., 2025). Several high-quality, human-annotated datasets have been curated, including ProofNet (Azerbayev et al., 2022), MiniF2F (Zheng et al., 2022), Putnam (Tsoukalas et al., 2024), and the Mathlib library itself (mathlib Community, 2020). While invaluable, creating these datasets is resource-intensive, requires domain experts, and is consequently limited in scale.

Finally, given data scarcity, previous work explored techniques for efficient usage of available data resources. Related methods aim to extract the inherent relationships between NL and FL in the data, proposing alignment methods based on symbolic equivalence and semantic consistency (Li et al., 2024). However, aligning NL and FL remains challenging when their formats and structures differ, making it difficult to transfer the LLMs knowledge and reasoning capabilities between NL and FL.

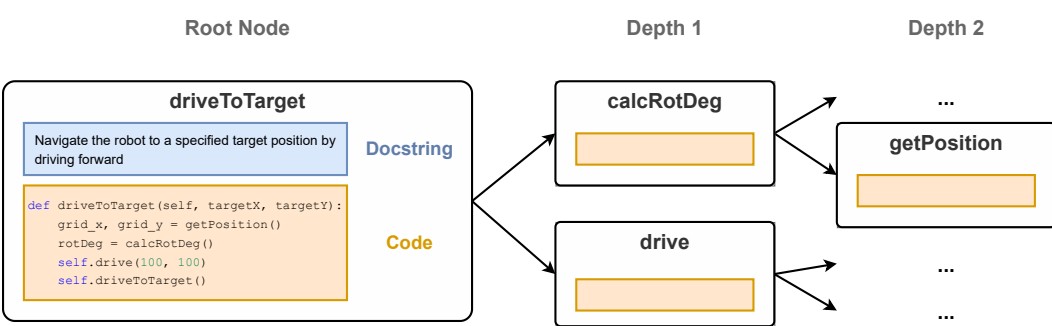

Figure 2: Function-level dependency tree showing the hierarchy of function calls, starting from the root node. Each child node represents a function called by its parent. The docstring for the root node is extracted to represent the description of the problem addressed in the code.

## 3 METHODOLOGY

We posit that current Math LLM performance is constrained primarily by the scarcity of large-scale training corpora. Large code datasets have already proven valuable for initializing these models (Shao et al., 2024; Wang et al., 2025), yet code data remains largely untapped during subsequent training on formal mathematics. Our TopoAlign framework and code autoformalisation task address this gap, demonstrating that structurally aligned code provides a complementary data source for mathematical autoformalisation.

### 3.1 TOPOLOGICAL DECOMPOSITION OF CODE FOR STRUCTURAL ALIGNMENT WITH FORMAL MATH DATA

TopoAlign builds on the premise that code and formal mathematical statements share a compositional structure. Functions in code solve distinct subproblems and may rely on auxiliary functions, analogous to how formal statements resolve informal problem descriptions using lemmata and theorems. We therefore decompose code at the function level (see Figure 1) into three transferable components: (i) the docstring, corresponding to the informal problem statement, (ii) the main function body, serving as a proxy for the formal statement; and (iii) its dependency functions, corresponding to supporting lemmata or library theorems (i.e., from Mathlib in Lean 4). We select Lean as the representative mathematical formal language in this work, while other systems such as Isabelle (Nipkow et al., 2002a) and Coq (The Coq Development Team, 2020) are discussed in Appendix E. Disassembling code into components that mirror those in formal mathematics enables structural transfer, allowing the aligned sequences to be used for training tasks such as autoformalisation and theorem proving. Importantly, TopoAlign targets structural alignment and does not assume semantic equivalence between programming-language type systems and Lean's dependent type system. The framework provides complementary pretraining data that enhances, rather than replaces, training on formal mathematical corpora.

To extract functional dependencies, we employ a topological dependency parser that performs a breadth-first search to build function-level dependency graphs (see Algorithm 1 in Appendix D). This contrasts with file-level dependency extraction (i.e., DeepSeek (Guo et al., 2024)), which captures inter-file execution order but omits intra-file functional relationships. Our parser leverages abstract syntax trees to trace calls and parent definitions across files, and is designed to handle standalone functions, class or instance methods, recursive calls, and imports. The result is a tree-structured representation of code dependencies, as illustrated in Figure 2.

To obtain informal problem statements analogous to those in mathematics, we extract natural language descriptions from docstrings and README files. However, standard docstring conventions do not always align with the needs of autoformalisation. Docstrings are often designed to describe a function's interface, including its inputs, outputs, and usage examples, rather than it's implementation, which is closer to the role of an informal mathematical statement. A summary of the function's implementation is therefore a more fitting analogue. Consequently, to create more suitable informal

descriptions and to augment missing or low-quality documentation, we generate concise summaries of each main function's logic using Qwen3 (Yang et al., 2025). We perform a 10-gram contamination analysis following Guo et al. (2024) to ensure no information leakage between our dataset and the test set, and confirm no meaningful overlap exists.[1]

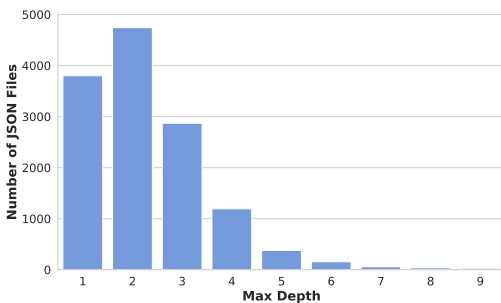
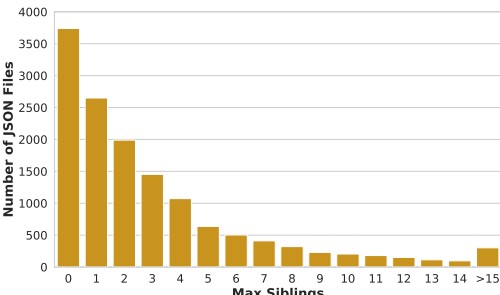

Figure 3: Distribution of maximum dependency tree depths across a random sample of 200 repositories.

Figure 4: Distribution of the maximum number of siblings in dependency trees across a random sample of 200 repositories.

We analyse the structural properties of function-level dependency trees in the collected code to select repositories whose hierarchical patterns resemble those found in formal mathematics. For each repository we compute the maximum tree depth, reflecting overall complexity and the maximum number of sibling nodes at any depth, indicating the breadth of direct dependencies. We retain repositories with depth between 3 and 6 and maximum sibling counts between 3 and 10, thus excluding overly complex codebases as well as simple scripts. Figures 3 and 4 show the distributions of maximum depth and maximum sibling count for a random sample of 200 repositories. After filtering, the corpus contains 156,684 functions comprising 324.5 million tokens.[2]

### 3.2 CODE AUTOFORMALISATION (CAF)

To leverage structurally aligned code for training Math LLMs on autoformalisation, we introduce code autoformalisation (CAF), which emulates the autoformalisation process on code data. Treating each aligned code instance as an analogue of a formalisation scenario allows the model to learn structural patterns while transferring general problem-solving strategies acquired in programming. Importantly, our method focuses on transferring structural and problem-solving capabilities, rather than introducing new mathematical knowledge. While we recognize that this technique could be adapted to emulate other formal reasoning tasks like theorem proving, in this work, we focus specifically on addressing the prevalent data scarcity bottleneck in autoformalisation.

We train Math LLMs using a mixture of TopoAlign code data and formal mathematical data. This combined approach integrates mathematical knowledge and structure with problem-solving capabilities from code, while also mitigating catastrophic forgetting during fine-tuning (Chen et al., 2019). In our multi-task training approach, each training sample consists of an input $x$, a docstring for code or an informal statement for math, and a set of dependencies $d$, a set of dependency functions for code or supporting lemmata and theorems for math. The model is conditioned on $x$ and $d$ and trained to generate the target $y$: the main function for code data or the formal statement for math data, as illustrated in Figure 5.

The proportion of code and math samples during training is controlled by parameter $\alpha$, where $\alpha$ determines the fraction of math samples and $1-\alpha$ the fraction of code samples. The overall objective is defined as $\mathcal{L} = \alpha\,\mathcal{L}_{\text{math}} + (1-\alpha)\,\mathcal{L}_{\text{CAF}}$, where $\mathcal{L}_{\text{math}}$ and $\mathcal{L}_{\text{CAF}}$ denote the losses for math and code tasks, respectively. For each task, the loss is computed using next-token prediction, formulated as the negative log-likelihood $\mathcal{L} = -\sum_{i=1}^{N} \log P_\theta\big(y_i \mid y_{<i}, x\big)$, where $x$ is the input sequence, $y = (y_1, y_2, \ldots, y_N)$ is the target sequence, and $P_\theta$ is the model with parameters $\theta$.

---

[1]Additional details are provided in Appendix C.

[2]Dataset available at: ANONYMIZED.

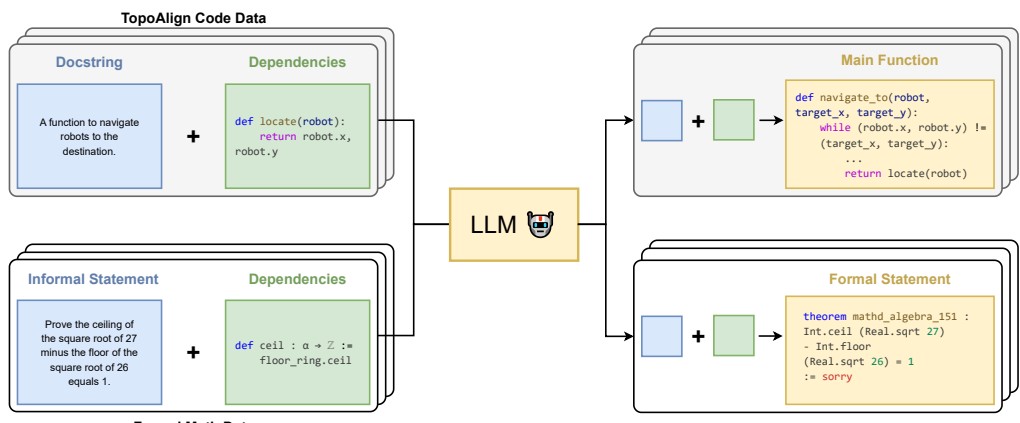

Figure 5: Overview of the training pipeline. The model takes a problem description (for either code or math) and its dependencies as input. The training objective is to generate the corresponding solution: the root code block for code inputs, or the formal statement for math inputs.

## 4 EXPERIMENTS

**Training Data.** Our code data is sourced from Python repositories in the Stack v2 dataset (Lozhkov et al., 2024). For formal mathematical statements, we use the Herald Statements corpus (Gao et al., 2025), a synthetic dataset built from Mathlib (mathlib Community, 2020). It contains Lean statements similar in style and structure to those found in math pretraining corpora, while avoiding overlap with downstream evaluation benchmarks such as MiniF2F, ProofNet, and Putnam. Dependency functions for the formal statements are extracted using the jixia library[3].

**Models.** We evaluate two base models: DEEPSEEK-MATH (Shao et al., 2024) and HERALD (Gao et al., 2025), each comprising 7 billion parameters. HERALD is specialised for autoformalisation, having been trained on the synthetic Herald Statements data. In contrast, DEEPSEEK-MATH is trained on a broader range of mathematical data from DeepSeek-Coder (Guo et al., 2024), optimised for general mathematical problem-solving rather than explicit autoformalisation. This setup allows us to compare the performance of a dedicated autoformalisation model against one with enhanced code understanding.

**Settings.** For each base model, we evaluate four distinct variations, namely **Baseline**, **Math**, **Code** and **TopoAlign**. The **Baseline** setting evaluates the pretrained models directly on downstream tasks without any additional fine-tuning. The **Math** setting involves fine-tuning models exclusively on formal mathematical data from the Herald Statements corpus, which is equivalent to applying the CAF objective with a mixing ratio of $\alpha = 1$; this allows us to assess the impact of training on purely mathematical data. In the **Code** setting, models are trained only on unaligned code data extracted from the Stack v2 corpus, providing a control to test the effect of training without structural alignment via TopoAlign. Finally, the **TopoAlign** setting trains models on a balanced combination of formal mathematics and structurally aligned code using CAF, with a mixing ratio of $\alpha = 0.5$. This setting uses approximately randomly selected 4,000 samples from GitHub repositories and 4,000 samples from the Herald Statements dataset. For consistency, the number of training samples is kept equal across all settings. Details on training hyperparameters and prompts are provided in Appendix A.

**Benchmarks.** We evaluate models on several Lean 4 autoformalisation benchmarks: Putnam (Tsoukalas et al., 2024), the validation and test sets of MiniF2F (Zheng et al., 2022), and ProofNet (Azerbayev et al., 2022). Each benchmark consists of paired natural language and formal language statements, with the autoformalisation task consisting of generating the correct formal statement

---

[3]https://github.com/frenzymath/jixia

| Dataset | Base Model | Setting | TC@1 | BEq@1 | TC@10 | BEq@10 |
|---|---|---|---|---|---|---|
| **MiniF2F-valid** | DEEPSEEK-MATH | Baseline | 0.00 | 0.00 | 0.00 | 0.00 |
| | | Math | 45.18 | 9.65 | 79.82 | 19.74 |
| | | Code | 42.98 | 10.53 | **88.16** | 23.68 |
| | | TopoAlign | **51.75** | **14.47** | 87.28 | **26.32** |
| | HERALD | Baseline | 73.25 | 25.44 | 92.98 | 38.16 |
| | | Math | 75.44 | 24.12 | 93.86 | **41.23** |
| | | Code | 75.00 | 20.61 | 90.79 | 32.02 |
| | | TopoAlign | **76.75** | **26.75** | **94.74** | **41.23** |
| **MiniF2F-test** | DEEPSEEK-MATH | Baseline | 0.00 | 0.00 | 0.00 | 0.00 |
| | | Math | 43.11 | 9.33 | 77.33 | 21.78 |
| | | Code | 52.89 | 7.56 | **91.52** | 26.79 |
| | | TopoAlign | **54.67** | **16.89** | 88.89 | **29.78** |
| | HERALD | Baseline | 78.22 | 24.44 | **95.54** | 41.96 |
| | | Math | **80.89** | 24.44 | **95.54** | 40.36 |
| | | Code | **80.89** | 20.89 | 90.63 | 34.38 |
| | | TopoAlign | 79.56 | **27.56** | 94.20 | **42.86** |
| **ProofNet** | DEEPSEEK-MATH | Baseline | 0.00 | 0.00 | 0.00 | 0.00 |
| | | Math | 21.12 | 5.35 | 43.42 | 12.57 |
| | | Code | 22.73 | 2.67 | 49.47 | 7.49 |
| | | TopoAlign | **32.89** | **9.09** | **56.95** | **14.97** |
| | HERALD | Baseline | 46.52 | 10.16 | 74.87 | **20.32** |
| | | Math | **46.79** | **10.43** | 75.40 | 19.77 |
| | | Code | 38.24 | 4.55 | 63.37 | 9.36 |
| | | TopoAlign | 43.85 | 9.63 | **75.67** | 16.84 |
| **Putnam** | DEEPSEEK-MATH | Baseline | 0.00 | 0.00 | 0.00 | 0.00 |
| | | Math | 10.69 | 0.00 | 27.04 | 0.00 |
| | | Code | 12.58 | 0.00 | 37.42 | 0.00 |
| | | TopoAlign | **17.30** | 0.00 | **42.14** | 0.00 |
| | HERALD | Baseline | 37.42 | 2.20 | 73.58 | **4.72** |
| | | Math | **43.71** | **2.52** | 70.44 | 4.09 |
| | | Code | 35.85 | 0.00 | 69.18 | 0.00 |
| | | TopoAlign | 36.16 | 1.57 | **76.73** | **4.72** |

Table 1: Auto-formalisation performance in percent for MiniF2F, ProofNet, and Putnam datasets under pass@1 and pass@10 metrics for Typecheck (TC) and BEq. Baseline setting refers to the pretrained model, Math is trained on additional formal math data and code is trained on additional code data that is not structurally aligned. Topoalign mixes math and structurally aligned code data.

given an informal description and its dependencies. Among these, Putnam presents the most challenging problems, ProofNet comprises textbook theorems, and MiniF2F is considered the most in-distribution benchmark, as it overlaps with Mathlib data.

**Evaluation Metrics.** We measure model performance using two primary metrics. First, we use Typecheck (TC) with the Lean 4 compiler (v4.11.0) to verify the syntactic correctness of generated statements (Poiroux et al., 2025; Limperg, 2025; Rabe et al., 2020; Wu et al., 2022b). For a more rigorous assessment of semantic fidelity, we employ bidirectional equivalence (BEq) (Liu et al., 2025a), which uses an LLM to generate proof tactics establishing logical equivalence between the model's output and the reference statement. This provides a stronger signal of faithful autoformalisation than typechecking alone. For both metrics, we report pass@k scores, where a sample is considered correct if at least one of its $k$ generated candidates passes the evaluation criterion.

## 5 RESULTS AND DISCUSSION

Table 1 presents the Typecheck and BEq performances for pass@k with $k = \{1, 10\}$, evaluated on the MiniF2F-valid, MiniF2F-test, ProofNet, and Putnam datasets.

Our proposed TopoAlign method consistently outperforms most of the baseline models in BEq across most datasets with substantial gains within both model families. For example, in the case of DEEPSEEK-MATH, the BEq@1 score on the MiniF2F-valid dataset increases from 9.65% to 14.47%. It is worth noting that BEq@1 can exhibit some variance due to the stochastic nature of sampling, particularly with temperature-based decoding. This variability explains occasional performance drops in certain cases such as the slightly lower BEq@1 observed for HERALD on the Putnam dataset. Additionally, in terms of Typecheck accuracy, our model demonstrates superior performance across all baselines as well with only a few exceptions observed among the HERALD variants. In the more robust BEq@10 evaluation, TopoAlign shows consistent and substantial improvements for both DEEPSEEK-MATH and HERALD across Putnam MiniF2F-valid and MiniF2F-test. These results demonstrate the effectiveness and generalisability of the TopoAlign in enhancing autoformalisation performance, particularly on less complex theorem formalisation datasets such as MiniF2F-valid and MiniF2F-test. These results suggest that the combination of topological alignment and code autoformalisation effectively transfers both problem-solving skills and structural knowledge from code to mathematical autoformalisation tasks.

Furthermore, TopoAlign consistently surpasses the code-only model variants for both the HERALD and DEEPSEEK-MATH models on both BEq and Typecheck metrics. This pattern implies that the underlying code dataset, when used without structural alignment, does not provide information beneficial for autoformalisation beyond basic problem-solving capabilities. In contrast, the topological alignment and CAF task enable the successful transfer of both advanced problem-solving and structural knowledge to math autoformalisation. Based on these findings, we conclude that TopoAlign and CAF effectively leverage code data to enhance the training of Math LLMs. This approach demonstrates that structurally aligned code datasets can serve as valuable sources of training data for Math LLMs, thereby addressing the scarcity of math-specific data. Principally, TopoAlign unlocks code repositories as structurally aligned pretraining data while leaving the underlying code content unchanged, and CAF then facilitates the transfer of this knowledge from code to mathematics. Our results validate that integrating widely available code data into the pretraining corpus substantially improves math autoformalisation performance, and even provides additional performance benefits for specialized autoformalization models.

## 5.1 QUALITATIVE ERROR ANALYSIS

To better understand the model's behavior, we conduct a qualitative error analysis on 40 randomly selected samples from the ProofNet dataset. Our analysis focuses on the pass@1 results for the HERALD model trained with TopoAlign. The HERALD + TopoAlign model successfully formalises 36 of the 40 samples according to BEq. Interestingly, when comparing these outputs to those from the HERALD + Math model, we observe that their successes are complementary: each model correctly formalises a distinct set of problems that the other fails on. Upon closer examination, we find that the HERALD + TopoAlign model typically generates the main semantic components correctly with respect to the ground truth. However, a frequent source of error is the incorrect assignment of variable types. For example, consider the problem:

Informal Statement: For all odd $n$, show that $8 \mid n^2 - 1$.

Autoformalisation (HERALD + TopoAlign):

```
theorem eigh_dvd_sq_sub_one_of_odd {n : Z}: Odd n → 8 | n^2 - 1 :=
    sorry
```

Ground Truth:

```
theorem exercise_1_27 {n : ℕ} (hn : Odd n) : 8 | (n^2 - 1) :=
    sorry
```

The crucial difference here is that the generated sample uses $n$ as an integer ($\mathbb{Z}$), whereas the ground truth requires $n$ to be a natural number ($\mathbb{N}$). This distinction is important, as $n^2 - 1$ must be non-

| Dataset | Data ratio | TC@1 | BEq@1 |
|---|---|---|---|
| Putnam | $\alpha = 0.25$ | 12.58 | 0.00 |
| | $\alpha = 0.50$ | **17.30** | 0.00 |
| | $\alpha = 0.75$ | 12.89 | 0.00 |
| ProofNet | $\alpha = 0.25$ | 24.60 | 6.15 |
| | $\alpha = 0.50$ | **32.89** | **9.09** |
| | $\alpha = 0.75$ | 26.20 | 8.29 |
| MiniF2F-valid | $\alpha = 0.25$ | 44.74 | 11.84 |
| | $\alpha = 0.50$ | 51.75 | **14.47** |
| | $\alpha = 0.75$ | **52.63** | 10.96 |
| MiniF2F-test | $\alpha = 0.25$ | 42.67 | 7.56 |
| | $\alpha = 0.50$ | 54.67 | **16.89** |
| | $\alpha = 0.75$ | **56.44** | 14.67 |

Table 2: Ablation study on training data composition for DEEPSEEK-MATH. The table compares Typecheck (TC) and bidirectional equivalence (BEq) scores for pass@1 to identify the optimal ratio of formal math data to our aligned code data.

negative. We hypothesize that this type of mismatch arises because the structurally aligned code data does not sufficiently emphasize type constraints. TopoAlign focuses on structural alignment, leaving fine-grained type distinctions as future work. One promising direction is incorporating strongly typed languages (e.g., Java, C++) during pretraining to mitigate the "type blindness" observed with weakly typed Python alone. Such type awareness would complement formal mathematics training and potentially improve correctness in autoformalization.

## 5.2 FINE-TUNING MATH MODELS ON CODE-ONLY DATA

We evaluated the HERALD model trained solely on code data to assess its ability to generalize to mathematical tasks. This experiment was motivated by the previously observed limitation of the DEEPSEEK-MATH model, which performs poorly on such tasks without targeted fine-tuning. The results show that the code-only HERALD model drops in performance, with BEq scores falling to zero and a significant decrease in typecheck accuracy. The generated outputs, while sometimes semantically plausible, are frequently syntactically invalid in Python. Representative examples of these outputs are provided in Appendix B.

## 5.3 IMPACT OF CODE–MATH DATA RATIO

We further explored the effect of varying the ratio of code to mathematics data in the CAF objective by adjusting the $\alpha$ parameter. To do this, we trained additional DEEPSEEK-MATH models with $\alpha$ values of 0.25 and 0.75, and report the results in Table 2. The balanced ratio ($\alpha = 0.5$) consistently yields the highest or near-highest performance for both Typecheck@1 and BEq@1. Lowering the mathematical content ($\alpha = 0.25$) leads to the weakest results, likely due to the token-level loss being dominated by the larger code samples, biasing the model toward code generation. Increasing the proportion of mathematical data ($\alpha = 0.75$) improves Typecheck@1 scores, especially on benchmarks closer to Mathlib, such as MiniF2F, but does not consistently improve BEq@1. These findings indicate that a balanced mix of code and mathematical data is crucial for optimal autoformalisation performance: mathematical data enhances syntactic accuracy, while code data enhances problem-solving capabilities.

## 5.4 BASE MODEL PERFORMANCE

The DEEPSEEK-MATH base model, which lacks pretraining on autoformalisation tasks, fails to generate meaningful outputs, often producing repeated symbols or malformed syntax that result in typecheck failures. In contrast, HERALD, which is pretrained on a large mathematical corpus, provides strong BEq performance across multiple datasets. Notably, the introduction of CAF training further improves its results: for instance, BEq@1 on MiniF2F-valid increases from 25.44% to 26.75%, and

on MiniF2F-test from 24.44% to 27.56%. These improvements confirm that TopoAlign provides significant benefits even for models that already possess strong autoformalisation capabilities.

## 6 CONCLUSION

This work shows that widely available code repositories are a valuable, previously untapped resource for pretraining more capable Math LLMs. We address the challenge of mathematical data scarcity by introducing TopoAlign, a method for structurally aligning code with formal mathematics through topological decomposition of docstrings, main functions, and dependency functions. Using this approach, we curated a 324.5 million token dataset that mirrors the structure of formal mathematical statements. Training DEEPSEEK-MATH and HERALD models on this dataset leads to substantial improvements across four autoformalisation benchmarks, as evidenced by gains in both Typecheck and BEq metrics. Our methodology successfully transfers structural and problem-solving knowledge from code to mathematical reasoning. Ablation studies further highlight the need for a balanced mix of code and formal mathematical data: code improves problem-solving ability, while mathematical data ensures syntactic accuracy. Our findings establish structurally aligned code as a resource for advancing Math LLMs and open new opportunities for scaling their capabilities by leveraging code repositories. Beyond autoformalization, TopoAlign's structural alignment may extend to other mathematical reasoning tasks such as proof generation, though the distinct structures of formal statements and proofs warrant careful investigation in future work.

### ETHICS STATEMENT

This research has been conducted in accordance with ethical standards and guidelines. The study does not involve human participants, animals, or sensitive data, and thus does not raise any ethical concerns. All data used in this research were obtained from publicly available sources and were handled in compliance with relevant data protection regulations. We affirm that the work presented in this paper adheres to the principles of integrity, transparency, and academic honesty. We used LLMs solely to assist in editing and improving the language of this manuscript.

### REPRODUCIBILITY STATEMENT

To ensure full reproducibility, all data sources, tools, and code are publicly available. The downstream evaluation benchmarks were sourced directly from their official repositories: MiniF2F (https://github.com/openai/miniF2F), Putnam (https://github.com/trishullab/PutnamBench), and Herald (https://huggingface.co/datasets/FrenzyMath/Herald_statements). Ground truth dependencies were processed using Jixia (https://github.com/frenzymath/jixia), as detailed in Section 4. Our complete codebase and the generated dataset have been released at ANONYMIZED.

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

## A    TRAINING HYPERPARAMETERS

The hyperparameters used for training the LLMs are listed in Tables 3 and 4. We adhere to the default settings provided in the original papers.

## B    GENERATED SAMPLE FROM CODE-ONLY TRAINED MODEL

These samples are generated by HERALD trained exclusively on code data. The outputs demonstrate that while the model captures the underlying logical structure correctly, it lacks proficiency in Lean 4 syntax. However, this suggests the presence of semantic similarity between programming code and formal mathematical language.

| Hyperparameters | DEEPSEEK-MATH | HERALD |
|---|---|---|
| learning rate | 2e-4 | 1e-6 |
| weight decay | 0.1 | 0.1 |
| epochs | 1 | 1 |
| batch size | 32 | 32 |

Table 3: Training hyperparameter settings.

| Hyperparameters | DEEPSEEK-MATH | HERALD |
|---|---|---|
| temperature | 0.70 | 0.70 |
| top p | 0.95 | 0.95 |
| max token | 1024 | 1024 |

Table 4: Inference hyperparameter settings used for autoformalisation in the pass@10 setting.

```python
from math import sqrt, ceil, floor

def evaluate_sqrt_expressions_1():
    assert(int(ceil(sqrt(27))) - int(floor(sqrt(26))) == 1)

def is_increasing_function(a, b):
    '''
    The function checks if the function f(x) = 4bx + (a+1)x^2 is
    increasing for x >= 0.
    '''
    return 4*b <= 4*b**2 + (a+1)**2

def is_3_(girls: list[int]) -> int:
    """
    >>> is_3_([1,2,3,4,5,6,7])
    3
    """
    return 3
```

## C   TRAINING INSTRUCTIONS AND INFERENCE PROMPTS

To generate docstrings describing the main functions in the code repositories, we employ Qwen3-14B model (`https://huggingface.co/Qwen/Qwen3-14B`) using the following prompt format:

**Prompt**

Provide a concise description of the problem solved in the code snippet below. Format the response as a docstring.
{code}

We manually verified 50 randomly sampled docstrings from our dataset, and all of them exhibit accurate and complete descriptions of the corresponding code. Selected samples are provided below:

```
756
757  def sync(self, title: str, keywords: Union[str, list[str]]) -> dict:
758
759      title = validate_title(title)
760      keywords = validate_keywords(keywords)
761      filter_item = self.filter(title)
762      remote_keywords = filter_item["keywords"]
763      [...]
764      params = OrderedDict(
765          {
766              "title": title,
767              "context[]": filter_item["context"],
768              "filter_action": filter_item["filter_action"],
769          }
770      )
771      params.update(self._build_keyword_params(add_keywords +
772      delete_keywords))
773      response = self._call_api(
774          "put", f"/api/v2/filters/{filter_item['id']}", params=params
775      )
776      response["added"] = add_keywords
777      response["deleted"] = delete_keywords
778      return response
```

**Generated Docstring**

This function synchronizes local and remote keywords for a filter by comparing provided keywords with existing ones. It identifies keywords to add and remove, logs changes, constructs API parameters, and updates the filter via a PUT request. The response includes added/deleted keywords to confirm synchronization.

```
790  def attention_mlp_stack(inputs, out_channel, num_heads, depth,
791      key_dim, attn_ratio, mlp_ratio, strides, stack_drop=0,
792      activation="hard_swish", name=""):
793
794      nn = inputs
795      embed_dim = nn.shape[-1]
796      stack_drop_s, stack_drop_e = stack_drop if isinstance(stack_drop,
797      (list, tuple)) else [stack_drop, stack_drop]
798      for ii in range(depth):
799          block_name = name + "block{}_".format(ii + 1)
800          drop_rate = stack_drop_s + (stack_drop_e - stack_drop_s) * ii
801      / depth
802          nn = res_mhsa_with_multi_head_position(nn, embed_dim,
803      num_heads, key_dim, attn_ratio, drop_rate, activation=activation,
804      name=block_name)
805
806      if mlp_ratio > 0:
807      [...]
808      if mlp_ratio > 0:
809          nn = res_mlp_block(nn, mlp_ratio, drop_rate,
      activation=activation, name=block_name + "mlp_")

      return keras.layers.Activation("linear", name=name + "output")(nn)
```

> **Generated Docstring**
>
> This function constructs a neural network stack with multi-head attention and MLP blocks. It applies a series of attention mechanisms (res_mhsa_with_multi_head_position) and optional MLPs (res_mlp_block) across 'depth' layers, adjusting dropout rates progressively. The stack supports configurable heads, key dimensions, ratios, and activation functions, returning a linear output layer.

These examples demonstrate that generated docstrings faithfully capture the code: the first enhances a minimal existing docstring with concise descriptions of keyword synchronization logic, while the second provides a detailed description of the multi-layer attention–MLP architecture for a function that originally lacked a docstring.

For math-related tasks, we use the following prompt format:

> **Prompt**
>
> Use the following pre-defined Lean 4 dependencies:
> {dependencies}
>
> Based on the context and the problem description, generate a single, syntactically correct Lean 4 formal statement that accurately captures the problem's meaning.
>
> Problem Description:
> {problem description}

For code-related tasks, we employ a parallel prompt structure:

> **Prompt**
>
> Use the following pre-defined functions:
> {pre-defined functions}
>
> Based on the context and the problem description, generate a syntactically correct function implementation that accurately captures the problem's meaning.
> Problem Description:
> {problem description}

## D    DEPENDENCY TREE EXTRACTION ALGORITHM

The dependency tree extraction process is formally described in Algorithm 1. The algorithm iterates through all files within a given directory, systematically identifying user-defined functions, including both class methods and standalone functions. We then analyse inter-file and inter-class dependency relationships. Through static analysis of function call patterns, we construct a directed dependency graph where each function call establishes a parent-child relationship. The calling function serves as the parent node, while the called function becomes the child node.

## E    EXTENDING TOPOALIGN TO OTHER FORMAL LANGUAGES

Formal statements in other proof assistants (e.g., Coq and Isabelle) share structural similarities with Lean. Below we show Isabelle and Coq versions of the informal statement: *Prove the ceiling of the square root of 27 minus the floor of the square root of 26 equals 1.*

> Isabelle:
> theorem ceil_sqrt27_minus_floor_sqrt26: "ceiling (sqrt 27) - floor (sqrt 26) = (1::int)"

**Algorithm 1** Function Call Dependency Analysis

---

**Require:** Python project directory
**Ensure:** Dependency graph of function calls
 1: **Data Structures:**
 2: $function\_definitions \leftarrow \{\}$            ▷ Record defined functions
 3: $imports \leftarrow \{\}$ $object\_types \leftarrow \{\}$     ▷ Imported names → full paths; Object names → class names; Save these two to track their parent function definitions or the file where they're defined
 4: ANALYZEFILE($f$) for all Python files $f$ in directory (recursively)
 5: **procedure** ANALYZEFILE(file_path)
 6:      Parse AST from $file\_path$
 7:      Initialize $function\_calls \leftarrow \{\}$, $object\_types \leftarrow \{\}$
 8:      **for all** nodes $n$ in AST **do**
 9:          **if** $n$ is **ClassDef then**
10:              Track current class context             ▷ Class defined function
11:          **else if** $n$ is **Import/ImportFrom then**
12:              Record $imports[alias] \leftarrow$ full module path     ▷ Function from other files
13:          **else if** $n$ is **FunctionDef then**
14:              Register function with $current\_class.name$ if applicable
15:              Initialize $function\_calls[name] \leftarrow []$          ▷ Save its dependency
16:          **else if** $n$ is **Assign** with constructor call **then**
17:              Map $object\_types[var] \leftarrow$ class name ▷ Record the codes that initialise this function
18:          **else if** $n$ is **Call then**             ▷ Record the codes that call this function
19:              Resolve full function name (direct call($obj()$), method call($obj.method()$), or imports $imports[name]$
20:              Append to $function\_calls[current\_func]$     ▷ The list to record the dependency information
21:          **end if**
22:      **end for**
23: **end procedure**
24: **procedure** BUILDDEPENDENCYGRAPH
25:      Filter to keep only project-internal calls
26:      Detect recursive calls ($f \rightarrow f$)             ▷ Self-recursive call
27:      **Topological Sort:**             ▷ BFS search
28:      Construct nested call tree from sorted order
29:      Insert recursion markers at tree root if needed
30: **end procedure**

---

> Coq:
> Theorem ceil_sqrt27_minus_floor_sqrt26 : (up (sqrt 27) - Z.floor (sqrt 26))%Z = 1%Z.

While these formal languages show similar structural characteristics to the decomposed code data in TopoAlign, supporting multiple formal languages would require either training dedicated models for each target language or constructing a mixed pretraining corpus that incorporates multiple formal languages. Although this direction offers promising practical applications, it lies beyond the scope of the present work.