# OpenReview forum: "TopoAlign: A Framework for Aligning Code to Math via Topological Decomposition"
_ICLR.cc/2026/Conference — Submitted to ICLR 2026_

### Official Review · Reviewer_TtGu · 2025-10-19

**Soundness:** 2
**Presentation:** 3
**Contribution:** 2
**Rating:** 6
**Confidence:** 3

**Summary:**

This paper proposes TopoAlign, a framework that decomposes code into docstrings, main functions, and dependency functions, and reassembles these components into analogues that structurally mirror formal statements. This produces structurally aligned code data that can be used for training Math LLMs without requiring additional human annotation.

**Strengths:**

1. The writing is clear and easy to follow
2. The idea of drawing a structural analogy to programming code to generate alignment data is novel, solving the bottleneck in training data.
3. The ablation study is thorough.

**Weaknesses:**

1. The code and data do not seem to be available, making reproduction difficult.
2. The choice of base model seems outdated. Perhaps training on newer Qwen models might better demonstrate the effectiveness of the training data.
3. Formal languages like Lean 4 are statically and dependently typed, making type correctness paramount. This represents a fundamental mismatch in the analogy the framework is built upon.

**Questions:**

See weaknesses.

---

> ### Author Response · Authors · 2025-11-21
> **Response to Reviewer TtGu**
>
> > The code and data do not seem to be available, making reproduction difficult.
>
> We are committed to open research and will release both the code and the TopoAlign data under the CC-BY license as soon as possible. This will enable full reproduction of our experiments and facilitate further work building on our framework.
>
> > The choice of base model seems outdated. Perhaps training on newer Qwen models might better demonstrate the effectiveness of the training data.
>
> The two base models we chose, DeepSeek-Math and Herald, were state-of-the-art Math LLMs for autoformalisation at the time of writing. To our knowledge, there is currently no dedicated Math LLM in the Qwen 3 family; however, we are happy to include supplementary experiments on the Qwen 3 Coder model to further demonstrate that TopoAlign's benefits are not tied to a specific backbone.
>
> > Formal languages like Lean 4 are statically and dependently typed, making type correctness paramount. This represents a fundamental mismatch in the analogy the framework is built upon.
>
> We respectfully disagree with the premise that our motivation relies on a fundamental mismatch between informal natural language and formal languages such as Lean 4. **Our framework does not assume that programming-language type systems are equivalent to Lean’s static, dependent type system. The core analogy in TopoAlign is compositional and structural**: we align docstrings, main functions, and their dependencies in code with informal statements, formal statements, and supporting lemmata in Lean. This concerns how problems are decomposed and solved, not the specific nature of the underlying type system.
>
> **We do not claim that training on TopoAlign can replace training on formal languages like Lean**. We believe that strongly typed languages can help mitigate the 'type blindness' observed when pretraining only on weakly typed Python (e.g., conflating integers and naturals), by encouraging the model to respect basic type constraints. This kind of general type awareness is complementary to, not a substitute for, exposure to logical type systems in formal mathematics.
>
> **Actions taken:** We have clarified this relationship between programming code and Lean 4, particularly with respect to types, in the revised manuscript in Sections 3.1 in line 197-201 and 5.1 in line 448-453.

---

> > ### Comment · Reviewer_TtGu · 2025-11-24
> >
> > Thank you for the clarification. I will keep my score and vote for acceptance of this paper.

---

### Official Review · Reviewer_EL9p · 2025-10-25

**Soundness:** 2
**Presentation:** 3
**Contribution:** 2
**Rating:** 4
**Confidence:** 3

**Summary:**

Autoformalisation from natural language to formal math (Lean 4) is bottlenecked by scarce parallel NL–FL data. This work decomposes code into (i) docstring → informal statement, (ii) main function → formal statement, and (iii) dependency functions → supporting lemmas; builds function-level dependency trees via AST/BFS; filters repositories by tree depth/breadth; augments or synthesizes docstrings via an LLM; introduces Code Autoformalisation (CAF) training mixing aligned code with formal math.

**Strengths:**

- Introduces a plausible structural bridge between code and formal math and operationalizes it at scale  (324.5M tokens).
- Demonstrates consistent improvements across multiple benchmarks and two model families, with meaningful gains for a non-specialized model (DeepSeek-Math).
- Thoughtful qualitative error analysis identifying type-related failure modes.

**Weaknesses:**

- no comparison to alternative structural alignments (e.g., file-level, call-graph without filtering), or to other synthetic data methods like ATLAS/student–teacher under the same budget.
- LLM-produced summaries may encode solution intent differently from natural informal statements; details missing.

**Questions:**

- Any contamination checks ensuring no overlap with evaluation sets (via code comments, problem text, or Lean entities)?
- How were the 4,000 code/math samples selected for TopoAlign runs? Random? Balanced by tree properties?

---

> ### Author Response · Authors · 2025-11-21
> **Response to Reviewer EL9p (1/2)**
>
> > no comparison to alternative structural alignments (e.g., file-level, call-graph without filtering), or to other synthetic data methods like ATLAS/student–teacher under the same budget.
>
> We firmly believe the **zero-shot, math-only, and code-only baselines presented in Table 1 are the most appropriate comparisons** for our method. Regarding the suggested alternatives: ATLAS generates fully synthetic mathematical problems and formalizations from existing Lean data for downstream specialization. **In contrast, TopoAlign utilizes general code repositories for pretraining, transferring structural knowledge and problem-solving patterns rather than Lean-specific mathematical content. These methods serve fundamentally different purposes (specialization vs. pretraining)** making direct comparison under equal budget less meaningful.
> Alternative structural alignments: File-level alignment would miss function-level dependencies critical to the "problem + subroutines" structure we aim to capture from formal mathematics. Using a full call graph without filtering is a reasonable variant; in our setup, the depth and branching thresholds are tunable, and we chose them to ensure that each instance includes sufficiently rich, non-trivial dependency structure while avoiding extremely noisy instances.
>
> **Actions taken:** We clarify these design choices and the distinction from ATLAS and related methods in the revised Section 2 in lines 132-134 and Section 4 lines 314-317.
>
> > LLM-produced summaries may encode solution intent differently from natural informal statements; details missing.
>
> We agree that many docstrings are incomplete or omit solution intent. **To address this concern, we use an LLM to generate more descriptive docstrings that mirror the level of detail found in informal problem descriptions in formal mathematics**. We provide two examples below:
>
> ```python
> def sync(self, title: str, keywords: Union[str, list[str]]) -> dict:
> """
> Sync filter.
> """
>     title = validate_title(title)
>     keywords = validate_keywords(keywords)
>     filter_item = self.filter(title)
>     remote_keywords = filter_item["keywords"]
>     [...]
>     params = OrderedDict(
>         {
>             "title": title,
>             "context[]": filter_item["context"],
>             "filter_action": filter_item["filter_action"],
>         }
>     )
>     params.update(self._build_keyword_params(add_keywords + delete_keywords))
>     response = self._call_api(
>         "put", f"/api/v2/filters/{filter_item['id']}", params=params
>     )
>     response["added"] = add_keywords
>     response["deleted"] = delete_keywords
>     return response
> ```
> **Generated Docstring:** This function synchronizes local and remote keywords for a filter by comparing provided keywords with existing ones. It identifies keywords to add and remove, logs changes, constructs API parameters, and updates the filter via a PUT request. The response includes added/deleted keywords to confirm synchronization.
>
> ```python
> def attention_mlp_stack(inputs, out_channel, num_heads, depth, key_dim, attn_ratio, mlp_ratio, strides, stack_drop=0, activation="hard_swish", name=""):
>     nn = inputs
>     embed_dim = nn.shape[-1]
>     stack_drop_s, stack_drop_e = stack_drop if isinstance(stack_drop, (list, tuple)) else [stack_drop, stack_drop]
>     for ii in range(depth):
>         block_name = name + "block{}_".format(ii + 1)
>         drop_rate = stack_drop_s + (stack_drop_e - stack_drop_s) * ii / depth
>         nn = res_mhsa_with_multi_head_position(nn, embed_dim, num_heads, key_dim, attn_ratio, drop_rate, activation=activation, name=block_name)
>     if mlp_ratio > 0:
>         [...]
>     if mlp_ratio > 0:
>         nn = res_mlp_block(nn, mlp_ratio, drop_rate, activation=activation, name=block_name + "mlp_")
>     return keras.layers.Activation("linear", name=name + "output")(nn)
> ```
>
> **Generated Docstring:** This function constructs a neural network stack with multi-head attention and MLP blocks. It applies a series of attention mechanisms (res_mhsa_with_multi_head_position) and optional MLPs (res_mlp_block) across 'depth' layers, adjusting dropout rates progressively. The stack supports configurable heads, key dimensions, ratios, and activation functions, returning a linear output layer.
>
> **Actions taken:** We included selected examples along with a qualitative analysis in Appendix C in line 756-822.

---

> > ### Author Response · Authors · 2025-11-21
> > **Response to Reviewer EL9p (2/2)**
> >
> > > Any contamination checks ensuring no overlap with evaluation sets (via code comments, problem text, or Lean entities)?
> >
> > We conducted contamination checks on the TopoAlign Code corpus using 10-gram overlap analysis, following the methodology of Guo et al. (2024) [1]. **Across all evaluation benchmarks, we found no information leakage**: zero instances in ProofNet and Putnam, and one instance in MiniF2F. Manual inspection of the single MiniF2F match revealed that the overlapping string "(p - 1) * (q - 1) * (r -" consists of common mathematical notation and does not leak problem-specific information. Given these results, we conclude that the performance benefits of training on TopoAlign are not due to information leakage.
> >
> > [1] Guo, D., Zhu, Q., Yang, D., Xie, Z., Dong, K., Zhang, W., … Liang, W. (2024). DeepSeek-Coder: When the Large Language Model Meets Programming -- The Rise of Code Intelligence. arXiv [Cs.SE]. Retrieved from http://arxiv.org/abs/2401.14196
> >
> > **Actions taken:** We include the results of the n-gram contamination analysis in Section 3.
> >
> > > How were the 4,000 code/math samples selected for TopoAlign runs? Random? Balanced by tree properties?
> >
> > The 4,000 samples from the code and math corpora were randomly selected.
> >
> > **Actions taken:** We improved our explanation of the sampling approach in Section 4 settings in line 328.

---

### Official Review · Reviewer_9age · 2025-10-30

**Soundness:** 2
**Presentation:** 3
**Contribution:** 2
**Rating:** 4
**Confidence:** 2

**Summary:**

This paper introduces TopoAlign, a framework that structurally aligns programming code with formal mathematical statements (e.g., in Lean 4) to address the scarcity of training data for autoformalization. By decomposing code into docstrings (informal statements), main functions (formal statements), and dependency functions (supporting lemmas), TopoAlign enables Math LLMs to learn compositional patterns from code. The authors also propose Code Autoformalisation (CAF), a training task that mimics autoformalization using aligned code. Experiments on benchmarks like MiniF2F, ProofNet, and Putnam show consistent improvements in both syntactic and semantic metrics.

**Strengths:**

* This paper leverages widely available code repositories as a scalable source of training data for formal mathematics, addressing a key bottleneck in autoformalization.

* The author conduct comprehensive experiments across multiple benchmarks and models, with clear ablation studies and qualitative analysis.

**Weaknesses:**

* The author assumes that the structure of an algorithm’s implementation (code topology) directly maps onto the logical and topological structure of the underlying mathematical theory. Real-world code often includes implementation noise (e.g., I/O, logging, dynamic memory management, exception handling) that are irrelevant to the formal mathematical concept and could introduce poor alignment.


* The alignment process heavily relies on the quality of the informal documentation (comments, docstrings, function names) within the codebase. If the code is poorly documented, the resulting informal side of the data pair will be low-quality or nonsensical.

* The method is evaluated primarily on autoformalization; its applicability to other reasoning tasks (e.g., theorem proving) is not fully explored.

* Improvements for HERALD are small (e.g., +1% BEq@10), suggesting diminishing returns for models already optimized for formalization.


* The following duplicate references should be merged into a single entry:
Zenan Li, Yifan Wu, Zhaoyu Li, Xinming Wei, Xian Zhang, Fan Yang, and Xiaoxing Ma. Autoformalize mathematical statements by symbolic equivalence and semantic consistency. Advances in Neural Information Processing Systems, 37:53598–53625, 2024a. Zenan Li, Yifan Wu, Zhaoyu Li, Xinming Wei, Xian Zhang, Fan Yang, and Xiaoxing Ma. Autoformalize mathematical statements by symbolic equivalence and semantic consistency. In Advances in Neural Information Processing Systems (NeurIPS) 2024, 2024b. URL https: //arxiv.org/abs/2410.20936. NeurIPS 2024 conference paper; code available online.

**Questions:**

* How does TopoAlign perform on out-of-distribution or non-Lean formal systems (e.g., Isabelle, Coq)?

* Is there any evidence that TopoAlign helps in downstream theorem proving (beyond autoformalisation)?

---

> ### Author Response · Authors · 2025-11-21
> **Response to Reviewer 9age (1/2)**
>
> > The author assumes that the structure of an algorithm’s implementation (code topology) directly maps onto the logical and topological structure of the underlying mathematical theory. Real-world code often includes implementation noise (e.g., I/O, logging, dynamic memory management, exception handling) that are irrelevant to the formal mathematical concept and could introduce poor alignment.
>
> We agree with the reviewer that real-world code contains implementation-specific noise that does not directly correspond to formal mathematics. **TopoAlign is explicitly designed to address structural, not semantic, alignment: it focuses on compositional patterns, rather than assuming a one-to-one mapping between every line of code and a mathematical concept**. Our goal is not to claim that arbitrary code perfectly reflects the topology of a formal mathematics, but that, after this topological decomposition and filtering, the resulting sequences capture compositional and problem-solving patterns that transfer well to formal mathematics. The empirical gains on both DeepSeek-Math and Herald, support that this structurally aligned subset of code provides useful signal rather than being dominated by implementation noise.
>
> > The alignment process heavily relies on the quality of the informal documentation (comments, docstrings, function names) within the codebase. If the code is poorly documented, the resulting informal side of the data pair will be low-quality or nonsensical.
>
> First, we would like to clarify that our framework relies on docstrings; comments or function names are not relevant for the structural alignment. We agree that relying on existing docstrings would make the alignment sensitive to documentation quality. To address this, we do not depend solely on human-written documentation. As described in Section 3.1, **for each main function we generate concise natural-language summaries with an LLM (Qwen 3)**. Docstrings typically describe a function’s interface (inputs, outputs, usage), which is not well aligned with the role of informal mathematical statements, which encode the content of the intended result. **Our LLM-generated summaries explicitly target the implementation-level behavior of the main function, making them a closer analogue to informal math problem statements.**
>
> **Actions taken:** We provided additional clarifications in the manuscript in Section 3.1 between line 215-220 and appendix C between 752-754.
>
> > The method is evaluated primarily on autoformalization; its applicability to other reasoning tasks (e.g., theorem proving) is not fully explored.
> > Q2: Is there any evidence that TopoAlign helps in downstream theorem proving (beyond autoformalisation)?
>
> We thank the reviewer for this insightful suggestion. Our work deliberately focuses on autoformalization as a first step, as it directly tests whether models can learn to map natural language to formal mathematical statements. **We agree that structural alignment principles could extend to theorem proving, but this requires careful adaptation**: formal statements and formal proofs have fundamentally different structures. While TopoAlign's functional dependencies may closely resemble the hierarchical structure of theorems and lemmas in whole-proof generation, applying the method to proof generation would require substantially different experimental setups and analysis. We view this as valuable future work.
>
> **Actions taken:** We discuss the usage of TopoAlign for theorem proving in the Conclusion Section in lines 517-520 .
>
> > Improvements for HERALD are small (e.g., +1% BEq@10), suggesting diminishing returns for models already optimized for formalization.
>
> We agree that the absolute BEq@10 gains for Herald are modest, which is expected given that Herald is already highly specialized and trained on the Herald statements dataset. **Our method is primarily designed to transfer structural and problem-solving capabilities rather than introduce new mathematical knowledge**. Therefore, training on TopoAlign data is most valuable at the pretraining stage. In this light, we see it as a positive result that, **even without adding any new mathematical content, code autoformalisation with TopoAlign still yields improvements in both BEq@10 and typecheck@10 for Herald**.
>
> **Actions taken:** We clarify the benefits of TopoAlign and CAF for Math LLM pretraining in Section 5, line 375-377.

---

> > ### Author Response · Authors · 2025-11-21
> > **Response to Reviewer 9age (2/2)**
> >
> > > The following duplicate references should be merged into a single entry: Zenan Li, Yifan Wu, Zhaoyu Li, Xinming Wei, Xian Zhang, Fan Yang, and Xiaoxing Ma. Autoformalize mathematical statements by symbolic equivalence and semantic consistency. Advances in Neural Information Processing Systems, 37:53598–53625, 2024a. Zenan Li, Yifan Wu, Zhaoyu Li, Xinming Wei, Xian Zhang, Fan Yang, and Xiaoxing Ma. Autoformalize mathematical statements by symbolic equivalence and semantic consistency. In Advances in Neural Information Processing Systems (NeurIPS) 2024, 2024b. URL https: //arxiv.org/abs/2410.20936. NeurIPS 2024 conference paper; code available online.
> >
> > **Actions taken:** Thank you for pointing this out. We have merged the duplicate entries into a single reference.
> >
> > > How does TopoAlign perform on out-of-distribution or non-Lean formal systems (e.g., Isabelle, Coq)?
> >
> > It is plausible that TopoAlign's code autoformalization data can be adapted to train models for different formal languages such as Isabelle or Coq, as these languages share similar structural properties in their formal statements. However, this would require either training separate models for each target language or incorporating multiple formal languages into a mixed pretraining corpus to develop a "multilingual" math model. While this presents interesting practical applications, we consider this extension beyond the scope of the current work.
> >
> > **Actions taken:** We provide a short discussion on the extension to other formal mathematical languages and training resources in the Appendix.

---

### Official Review · Reviewer_oAtz · 2025-11-02

**Soundness:** 3
**Presentation:** 3
**Contribution:** 3
**Rating:** 6
**Confidence:** 4

**Summary:**

This paper introduces TopoAlign, a framework to address data scarcity in mathematical autoformalization. It leverages code repositories by decomposing them into docstrings, main functions, and dependencies. This creates a large, structurally-aligned dataset. The authors use this for a CAF task, mixing aligned code with real math data. Experiments show this method improves generalist models (DEEPSEEK-MATH) and modestly improves specialist models (HERALD) by transferring structural and problem-solving skills from code.

**Strengths:**

Creatively uses abundant code repositories to solve the critical data bottleneck in formal mathematics.

The structural alignment between code and formal math is intuitive and well-justified.

Effectively demonstrates the method's value on DEEPSEEK-MATH, and includes a key ablation study on the optimal code-to-math data ratio.

**Weaknesses:**

As shown in Table 1, gains on the already specialized HERALD model are marginal, especially on MiniF2F-test, ProofNet and Putnam, suggesting the method is better for initializing models than pushing the state-of-the-art.

There is a fundamental semantic gap between its code data source and the target domain of formal mathematics. The model, trained on weakly-typed Python, fails to learn critical type distinctions, such as confusing a logical integer Z with a natural number N. The paper hypothesizes this could be fixed by using strongly-typed languages like Java or C++, but this solution may be flawed. It incorrectly equates computational type systems (like Java's int, for memory safety) with logical type systems (like Lean's Z, for abstract proof). The concepts a model would learn from Java are not the same as the required mathematical concepts. This represents a deep semantic gulf between programming paradigms and logical reasoning that simply changing the source language cannot bridge.

**Questions:**

none

---

> ### Author Response · Authors · 2025-11-21
> **Response to Reviewer oAtz**
>
> > As shown in Table 1, gains on the already specialized HERALD model are marginal, especially on MiniF2F-test, ProofNet and Putnam, suggesting the method is better for initializing models than pushing the state-of-the-art.
>
> While we agree with the reviewer that **our data is most valuable at the pretraining stage**, our experiments indicate that structurally aligned code can provide complementary benefits even for strong, task-specific autoformalisation models. Our method is **designed to transfer structural and problem-solving capabilities rather than introducing new mathematical knowledge**. The experiments on Herald confirm that, even without adding new mathematical knowledge (Herald is trained on the Herald statements dataset), code autoformalisation using TopoAlign data still yields typecheck and BEq performance improvements.
>
> **Actions taken:** We clarify the positioning of our method and dataset in the introduction section and results section in line 88-89 and 376-377.
>
> > There is a fundamental semantic gap between its code data source and the target domain of formal mathematics. The model, trained on weakly-typed Python, fails to learn critical type distinctions, such as confusing a logical integer Z with a natural number N. The paper hypothesizes this could be fixed by using strongly-typed languages like Java or C++, but this solution may be flawed. It incorrectly equates computational type systems (like Java's int, for memory safety) with logical type systems (like Lean's Z, for abstract proof). The concepts a model would learn from Java are not the same as the required mathematical concepts. This represents a deep semantic gulf between programming paradigms and logical reasoning that simply changing the source language cannot bridge.
>
> We agree that there is both a structural and a semantic gap between code and formal mathematics. TopoAlign is explicitly designed to address the structural gap: it reorganizes existing code to mirror the structure of formal mathematics. In this sense, **TopoAlign unlocks code repositories as structurally aligned pretraining data, while leaving the underlying code content unchanged**; the resulting data is semi-synthetic in structure but preserves the original semantics of both code and math. Our methods are not aiming to solve the entire semantic alignment issue.
> Regarding types, **we do not claim that training on strongly typed programming languages can replace training on formal systems like Lean, nor that Java's type system is conceptually equivalent to Lean’s logical types**. Our point is narrower: strongly typed languages can help mitigate the 'type blindness' observed when pretraining only on weakly typed Python (e.g., conflating integers and naturals), by encouraging the model to respect basic type constraints. This kind of general type awareness is complementary to, not a substitute for, exposure to logical type systems in formal mathematics.
>
> **Actions taken:** We clarify this in Section introduction and 5.1 Qualitative Error Analysis in line 448-453.
>
> We appreciate that the reviewer did not raise concerns about our methodology, and we have aimed to address all points in the reviews as clearly and thoroughly as possible. We therefore hope that the reviewer will consider these clarifications when reassessing the paper's overall evaluation.

---

### Author Response · Authors · 2025-12-02
**Rebuttal Summary and Discussion**

We thank the area chair and the reviewers for their time and feedback. Below, we summarize the key points discussed during the rebuttal and address overlapping comments across reviews.

**Clarification on aims for TopoAlign:**
- TopoAlign is explicitly designed to address the **structural gap between programming code and formal mathematical statements**: it reorganizes existing code repositories to mirror the structure of formal mathematics. In this sense, TopoAlign unlocks code repositories as structurally aligned pretraining data, while leaving the underlying code content unchanged.

**Performance gains for specialized autoformalizers, i.e. HERALD, are marginal:**
- We agree that the absolute BEq@10 gains for Herald are modest, which is expected given that TopoAlign does not introduce new mathematical knowledge, as Herald is already trained on the Herald statements dataset. Our method is primarily designed to transfer structural and problem-solving capabilities. Therefore, training on TopoAlign data is most valuable at the pretraining stage. In this light, we see it as a positive result that, even without adding any new mathematical content, code autoformalisation with TopoAlign still yields improvements in both BEq@10 and typecheck@10 for Herald.

**Dataset clarifications including data types in Lean 4 compared to programming languages, data contamination analysis, and Docstring generation:**
- We do not claim that training on strongly typed programming languages can replace training on formal systems like Lean, nor that Java's type system is conceptually equivalent to Lean’s logical types. We suggest that strongly typed languages can help mitigate the 'type blindness' observed when pretraining only on weakly typed Python (e.g., conflating integers and natural numbers), by encouraging the model to respect basic type constraints.
- We conducted an additional contamination analysis (see revised Section 3) on the TopoAlign Code corpus using 10-gram overlap analysis, following the methodology of Guo et al. (2024). Across all evaluation benchmarks, we found no information leakage. Given these results, we conclude that the performance benefits of training on TopoAlign are not due to information leakage.
- To mitigate sensitivity to documentation quality, we generate concise summaries for each main function using Qwen 3. Unlike typical docstrings describing interfaces, these summaries capture implementation-level behavior and the content is more closely related to informal mathematical statements.

**Additional experimental results using more recent Qwen 3 4B Instruct:**
- The two base models we chose, DeepSeek-Math and Herald, were state-of-the-art Math LLMs for autoformalisation at the time of writing. To the best of our knowledge, there is currently no dedicated Math LLM in the Qwen 3 family. However, we provide additional experimental results for Qwen 3 4B Instruct below and in the camera-ready manuscript. Qwen 3 models are pretrained on large-scale math corpora, including reasoning datasets generated using Qwen 2.5 Math and Coder models (around 5T tokens) (Qwen Team, 2025). The preliminary results on typecheck performance for Qwen 3 are in line with our previous findings and demonstrate that TopoAlign provides benefits independent of the backbone LLM.

| Dataset       | Base Model        | Setting   | TC@1  | TC@10  |
|---------------|-------------------|-----------|-------|--------|
| MiniF2F-valid | Qwen3-4B-Instruct | Baseline  | 2.63  | 21.93  |
|               |                   | TopoAlign | 77.19 | 100.00 |
| MiniF2F-test  | Qwen3-4B-Instruct | Baseline  | 6.22  | 37.33  |
|               |                   | TopoAlign | 88.89 | 100.00 |
| ProofNet      | Qwen3-4B-Instruct | Baseline  | 6.42  | 31.02  |
|               |                   | TopoAlign | 31.02 | 95.72  |
| Putnam        | Qwen3-4B-Instruct | Baseline  | 1.89  | 16.98  |
|               |                   | TopoAlign | 19.50 | 97.48  |

---

### Meta-Review · Area_Chair_9ekX · 2026-01-07

**Summary:**

This paper proposes TopoAlign, a framework to address the data scarcity in autoformalization by aligning programming code with formal mathematical statements. It decomposes code into docstrings, main functions, and dependency functions, and reassembles these components into analogues that structurally mirror formal statements. Experiments are conducted by training DeepSeek-Math and Herald, and then evaluating them on MiniF2F, Putnam, and ProofNet.

**Reviewer Concerns:**

Addressed reviewer concerns:
* Compared baselines seem outdated (Reviewer TtGu): Added experiments with Qwen3.
* Semantic gap between code and formal math types (Reviewer oAtz, TtGu): Authors clarified that this work focuses on structural.
* Reliance on documentation quality/noise in code (Reviewer 9age): Provided further explanation.
* Potential data contamination (Reviewer EL9p): Addressed with 10-gram overlap analysis showing no leakage.
* Lack of comparison to alternative alignment methods (Reviewer EL9p): Provided further explanation.

Unresolved reviewer concerns with insufficient experiments, supplementary materials, or explanations:
* Reproducibility (Reviewer TtGu)
* Limited gains for specialized models (Reviewers oAtz, 9age)
* Evaluation beyond autoformalization, including e.g., theorem proving (Reviewer 9age)

**Reviewer Scores:**

Based on the author response, while there is a possibility that reviewers might raise their scores, the overall ratings are likely not sufficiently high to secure the acceptance of this paper.

---

### Decision · Program_Chairs · 2026-01-26

Reject